# Synthesis, antibacterial activity, *in silico* ADMET prediction, docking, and molecular dynamics studies of substituted phenyl and furan ring containing thiazole Schiff base derivatives

**Md. Din Islam**[1], **Joyanta Kumar Saha**[2], **Sumita Saznin Marufa**[2], **Tanmoy Kumar Kundu**[2], **Ismail Hossain**[2], **Hiroshi Nishino**[3], **Mohammad Sayed Alam** (ID)[2]*, **Md. Aminul Haque**[2]*, **Mohammad Mostafizur Rahman** (ID)[2]*

**1** Department of Chemistry, Chittagong University of Engineering & Technology, Chattogram, Bangladesh, **2** Department of Chemistry, Jagannath University, Dhaka , Bangladesh, **3** Department of Chemistry, Graduate School of Science and Technology, Kumamoto University, Kumamoto, Japan

* msalam@chem.jnu.ac.bd (MSA); amin2k12@chem.jnu.ac.bd (MAH); mostafiz@chem.jnu.ac.bd (MMR)

## Abstract

This study synthesized eighteen phenyl and furan rings containing thiazole Schiff base derivatives **2(a–r)** in five series, and spectral analyses confirmed their structures. The *in vitro* antibacterial activities of the synthesized analogs against two gram-positive and two gram-negative bacteria were evaluated by disk diffusion technique. Compounds (**2d**) and (**2n**) produced prominently high zone of inhibition with 48.3 ± 0.6 mm and 45.3 ± 0.6 mm against *B. subtilis*, respectively, compared to standard ceftriaxone (20.0 ± 1.0 mm). However, the antibacterial potency of the compounds with furan ring was more notable than that of phenyl ring-containing derivatives. Molecular docking and dynamic study were performed based on the wet lab outcomes of (**2d**) and (**2n**), where both derivatives remained in the binding site of the receptors during the whole simulation time with RMSD and RMSF values below 2 nm. *In silico* ADMET prediction studies of the synthesized compounds validated their oral bioavailability. A more detailed study of the quantitative structure-activity relationship is required to predict structural modification on bioactivity and MD simulation to understand their therapeutic potential and pharmacokinetics.

## Introduction

The treatment of infectious diseases caused by various pathogens has become a global problem due to the extensive use of antibiotics and the rapid development of multidrug resistance against microorganisms. It was reported that the most commonly used antibiotics, such as fluconazole, chloramphenicol, penicillin, and amphotericin B, exhibited resistance against various microorganisms [1]. To overcome this problem, there is an acute need to develop new antimicrobial drugs with high potency. Electron-rich S and N atom-containing five-membered heterocyclic compounds with thiazole scaffold a key pharmacophore in

**Data availability statement:** "All relevant data are within the paper and its Supporting Information files."

**Funding:** Authors are thankful to the Ministry of Science and Technology (MOST), Bangladesh for funding in this project (Project number PHY'S 553, 2020-2021). The project was partially supported by Jagannath University, Dhaka, Bangladesh.

**Competing interests:** The authors have declared that no competing interests exist.

synthetic medicinal sectors that possess a wide range of bioactivities due to their low toxicity [2]. Numerous natural and synthesized compounds with thiazole skeleton showed anti-inflammatory [3], antifungal [4], antitubercular [5], antimicrobial [6], anticancer [7], antiviral [8], antischizophrenic [9], anti-HIV [10], antiplasmodial [11], and antioxidant [12], like medicinal properties. Moreover, 1,3-thiazole scaffold is commonly found in the chemical structure of many commercially available drugs such as vitamin B1 (thiamine), tiazofurin, dasatinib, ritonavir, meloxicam, and nitazoxanide [13–15]. The hydrazinylthiazolyl nucleus has drawn intense interest from organic synthetic chemists due to its prominent antibacterial and antioxidant activities [16–18]. Examples of several commercial drugs, containing thiazole ring are shown in **Fig 1**.

On the other hand, Schiff bases are the imine (or azomethine) group containing condensation products of primary amines and carbonyl compounds. They have achieved significant

**Fig 1. Structures of representative drugs having thiazole moieties. The thiazole skeletons are shown in red lines.**

importance in medicinal sectors due to their substantial antimicrobial, anticancer, anti-HIV, antitumor, and antioxidant properties [19–23]. Respective biological activities of the Schiff base compound might have appeared due to the presence of the imine group. The above-mentioned bioactivities could be modified by changing the substituents of the Schiff base molecules [24,25].

Due to the versatile bioactivity, furan pharmacophore has been used as a multifaceted structure for developing potential therapeutic agents [26]. Moreover, compounds containing furan moiety exhibited antimicrobial [27], antitumor [28], anti-oxidative [29], and antidepressant [30] activities. Efforts are continuing to design and develop highly potent and less toxic drugs to overcome the aforementioned problems. These days, it has become an efficient trend to use molecular hybridization for the development of new drugs by fusing two or more active pharmacophoric units into a single molecular structure. Considering the significant biological diversity of thiazoles and Schiff bases, the present study aimed to synthesize a series of substituted phenyl and furan moieties containing thiazole Schiff bases. It was reported that alkyl and nitro substituents attached to heterocyclic rings exhibited higher bioactivity against pathogenic diseases [31,32]. Our recent study also revealed that the alkyl-substituted heterocyclic and meta-substituted thiazole compounds increased the biological potential against microbial strains, supported by *in silico* studies [33–35]. The findings inspired us to synthesize a range of substituted-phenyl and -furan moieties containing thiazole Schiff bases for the broadened study of the related compounds. After synthesis, *in vitro* antimicrobial activities of the synthesized analogs were evaluated against gram-positive and gram-negative bacteria strains by disk diffusion method. To know the pharmacokinetic properties, (a) *in silico* toxicity, (b) drug-likeness, and (c) drug scores were calculated by Osiris property explorer. Molecular docking studies were executed to know the interaction of the synthesized compounds with the target protein receptors. Molecular dynamics studies (MD) further examined the docked complexes to verify their conformational changes and stability inside the respective protein receptor. This process performed calculations for root mean square deviations (RMSD), root mean square fluctuations (RMSF), and binding free energies.

## Experimental

### General methods

The synthesized compounds' melting points were estimated using the SMP10 apparatus and are uncorrected. FTIR spectra were recorded using samples running as KBr pellets by SHIMADZU IR Tracer-100 infrared spectrometer, and only major peaks were presented in cm$^{-1}$. $^1$H NMR spectra were recorded on a BRUKER 400 MHz NMR spectrometer at BCSIR laboratories, Dhaka, Bangladesh. Chemical shifts ($\delta$) were stated in parts per million (ppm) relative to TMS (internal reference standard), and coupling constants ($J$ values) were shown in Hertz (Hz). High-resolution mass spectra (HRMS) were measured at the analytical center of Kumamoto University, Japan. Reagents were obtained from Sigma-Aldrich and TCI Chemical Industries, Ltd (India) unless otherwise noted, and were used without further purification.

### General procedure for the synthesis of 2-{N-2-[(aryl)methylidene]hydrazin-1-yl}-1,3-thiazoles 2(a–l)

A mixture of thiosemicarbazide (0.005 mole) and substituted benzaldehydes (0.005 mole) or substituted acetophenones (0.005 mole) in 25.0 mL of ethanol was allowed to reflux at 80 °C. The reaction proceeded with continuous stirring until its completion. The Thin layer chromatography (TLC) method checked the progress of the reaction. After completion of the reaction, the mixture was cooled to room temperature and then filtered to collect crude products.

The crude products were recrystallized from ethanol to obtain the substituted benzaldehyde/acetophenone thiosemicarbazones **1(a–j)** (Scheme 1). In the following step, a mixture of substituted benzaldehyde/acetophenone thiosemicarbazones (0.002 mole) and chlorocarbonyl derivatives (0.002 mole) in 20.0 mL of acetone was refluxed (60 °C) with continuous stirring until the completion of the reaction. The reaction progress was monitored using the TLC technique. After resting, the reaction mixture was placed at room temperature, and the crude solid was separated by filtration. The filtrated crude was recrystallized from ethanol to yield final products **2(a–l)** (Schemes 2 and 3).

2-[N-2-(2-Hydroxyphenyl-2-ylmethylidene)hydrazine-1-yl]-4,5-dimethyl-1,3-thiazole (2a). Yellow solid, yield = 74.2% (0.367 g), m.p. = 115–117 °C. IR (KBr) $v$ (cm$^{-1}$): 3447 (NH), 3121 (O-H), 1627 (C=N), 1496 (C=C), 1114 (C-O), 761 (C-S-C). $^1$H NMR (400 MHz, CDCl$_3$) $\delta$ (ppm): 2.24 (s, 3H, CH$_3$), 2.26 (s, 3H, CH$_3$), 6.94–7.00 (m, 2H, Ar H), 7.28 (d, 1H, $J$ = 7.6 Hz, Ar H), 7.34 (t, 1H, $J$ = 7.6 Hz, Ar H), 8.52 (s, 1H, CH=N). HRMS [M + H] + calculated for C$_{12}$H$_{14}$N$_3$OS: ($m/z$) 248.0858, found 248.0859.

2-[N-2-(3-Nitrophenyl-2-ylmethylidene]hydrazine-1-yl]-4,5-dimethyl-1,3-thiazole (2b). Yellow solid, yield = 63.5% (0.351 g), m.p. = 242–243 °C. IR (KBr) $v$ (cm$^{-1}$): 3450 (NH), 1611 (C=N), 1521 (C=C), 1346 (N=O), 735 (C-S-C). $^1$H NMR (400 MHz, CDCl$_3$) $\delta$ (ppm): 1.26 (s, 3H, CH$_3$), 2.26 (s, 3H, CH$_3$), 7.61 (t, 1H, $J$ = 8.0 Hz, Ar H), 7.91 (d, 1H, $J$ = 8.0 Hz, Ar H), 8.26 (d, 1H, $J$ = 8.0 Hz, Ar H), 8.36 (s, 1H, Ar H), 8.57 (s, 1H, CH=N). HRMS [M + H] + calculated for C$_{12}$H$_{13}$N$_4$O$_2$S: ($m/z$) 277.0759, found 277.0797.

1-{4-Methyl-2-[N-2-[1-(3-bromophenyl)ethylidene]hydrazin-1-yl]-1,3-thiazol-5-yl} methane (2c). Colorless solid, yield = 70% (0.454 g), m.p. = 162–163 °C. IR (KBr) $v$ (cm$^{-1}$): 3511 (N-H), 1616 (C=N), 1559 (C=C), 754 (C-S-C). $^1$H NMR (400 MHz, CDCl$_3$) $\delta$ (ppm): 2.26 (3H, s, CH$_3$), 2.27 (3H, s, CH$_3$), 2.50 (3H, s, CH$_3$), 7.30 (d, 1H, $J$ = 8.0 Hz, Ar H), 7.56 (d, 1H, $J$ = 8.8 Hz, Ar H), 7.66 (d, 1H, $J$ = 8.0 Hz, Ar H), 7.92 (1H, s, Ar H). HRMS [M + H] + calculated for C$_{13}$H$_{15}$BrN$_3$S: ($m/z$) 324.0170, found 324.0158.

| Compd. | R$^1$ | R$^2$ | R$^3$ | R$^4$ | Compd. | R$^1$ | R$^2$ | R$^3$ | R$^4$ |
|--------|-------|-------|-------|-------|--------|-------|-------|-------|-------|
| a | H | OH | H | H | f | CH$_3$ | Cl | H | Cl |
| b | H | H | NO$_2$ | H | g | CH$_3$ | H | H | CH$_3$ |
| c | CH$_3$ | H | Br | H | h | CH$_3$ | H | H | OCH$_3$ |
| d | CH$_3$ | H | H | F | i | CH$_3$ | H | H | NH$_2$ |
| e | CH$_3$ | H | H | Br | j | CH$_3$ | H | H | NO$_2$ |

**Scheme 1. Synthesis of substituted benzaldehyde or acetophenone thiosemicarbazones 1(a–j). The Schiff base is shown in blue lines.**

**Scheme 2. Synthesis of aryl-thiazole Schiff base hybrids 2(a–d). The Schiff base and thiazole skeletons are shown in blue and red lines, respectively.**

1-{4-Methyl-2-[N-2-[1-(4-fluorophenyl)ethylidene]hydrazin-1-yl]-1,3-thiazol-5-yl} methane (2d). Colorless solid, yield = 71% (0.374 g), m.p. = 122–123 °C. IR (KBr) $v$ (cm$^{-1}$): 3393 (N-H), 1616 (C=N azomethine), 1577 (C=C), 668 (C-S-C). $^1$H NMR (400 MHz, DMSO-$d_6$) $\delta$ (ppm): 2.33 (s, 3H, CH$_3$), 2.35 (s, 3H, CH$_3$), 2.50 (s, 3H, N=C-CH$_3$), 7.27 (m, 2H, Ar H), 7.96 (m, 2H, Ar H). HRMS [M + H] + calculated for C$_{13}$H$_{15}$FN$_3$S: ($m/z$) 264.0971, found 264.0969.

1-{4-Methyl-2-[N-2-[1-(3-bromophenyl)ethylidene]hydrazin-1-yl]-1,3-thiazol-5-yl}ethan-1-one (2e). Yellow solid, yield = 86% (0.606 g), m.p. = 212–213 °C. IR (KBr) $v$ (cm$^{-1}$): 3443 (N-H), 1651 (C=O), 1613 (C=N), 1559 (C=C), 668 (C-S-C). $^1$H NMR (400 MHz, CDCl$_3$) $\delta$ (ppm): 2.47 (s, 3H, CH$_3$), 2.53 (s, 3H, N=C-CH$_3$), 2.68 (s, 3H, COCH$_3$), 7.29 (t, 1H, $J$ = 8.0 Hz, Ar H), 7.57 (d, 1H, $J$ = 8.0 Hz, Ar H), 7.70 (d, 1H, $J$ = 8.0 Hz, Ar H), 7.91 (s, 1H, Ar H). HRMS [M + H] + calculated for C$_{14}$H$_{15}$BrN$_3$OS: ($m/z$) 352.0119, found 352.0139.

1-{4-Methyl-2-[N-2-[1-(4-fluorophenyl)ethylidene]hydrazin-1-yl]-1,3-thiazol-5-yl}ethan-1-one (2f). Colorless solid, yield = 90% (0.524 g), m.p. = 201–202 °C. IR (KBr) $v$ (cm$^{-1}$): 3422 (N-H), 1654 (C=O), 1598 (C=N), 1574 (C=C), 668 (C-S-C). $^1$H NMR (400 MHz, DMSO-$d_6$) $\delta$ (ppm): 2.32 (s, 3H, CH$_3$), 2.39 (s, 3H, CH$_3$), 2.49 (s, 3H, N=C-CH$_3$), 7.24 (m, 2H, Ar H), 7.84 (m, 2H, Ar H). HRMS [M + H] + calculated for C$_{14}$H$_{15}$FN$_3$OS: ($m/z$) 292.0920, found 292.1007.

1-{4-Methyl-2-[N-2-[1-(4-bromophenyl)ethylidene]hydrazin-1-yl]-1,3-thiazol-5-yl}ethan-1-one (2g). Colorless solid, yield = 87% (0.613 g), m.p. = 241–243 °C. IR (KBr) $v$ (cm$^{-1}$): 3448 (N-H), 1653 (C=O), 1607 (C=N), 1585 (C=C), 668 (C-S-C). $^1$H NMR (400 MHz, CDCl$_3$): $\delta$ (ppm): 2.48 (s, 3H, CH$_3$), 2.53 (s, 3H, N=C-CH$_3$), 2.68 (s, 3H, COCH$_3$), 7.55 (d, 2H, $J$ = 8.4 Hz, Ar H), 7.66 (d, 2H, $J$ = 8.8 Hz, Ar H). HRMS [M + H] + calculated for C$_{14}$H$_{15}$BrN$_3$OS: ($m/z$) 352.0119, found 352.0132.

**Scheme 3. Synthesis of aryl-thiazole Schiff base hybrids 2(e–l). The Schiff base and thiazole skeletons are shown in blue and red lines, respectively.**

1-{4-Methyl-2-[N-2-[1-(2,4-dichlorophenyl)ethylidene]hydrazin-1-yl]-1,3-thiazol-5-yl} ethan-1-one (2h). Yellow solid, yield = 92% (0.630 g), m.p. = 221–222 °C. IR (KBr) $v$ (cm⁻¹): 3456 (N-H), 1649 (C=O), 1601 (C=N), 668 (C-S-C). ¹H NMR (400 MHz, CDCl₃): $\delta$ (ppm): 2.49 (s, 3H, CH₃), 2.50 (s, 3H, COCH₃), 2.68 (s, 3H, N=C-CH₃), 7.33-7.25 (m, 2H, Ar H), 7.45 (s, 1H, Ar H). HRMS [M + H] + calculated for $C_{14}H_{14}Cl_2N_3OS$: ($m/z$) 342.0235, found 342.0247.

1-{4-Methyl-2-[N-2-[1-(4-methylphenyl)ethylidene]hydrazin-1-yl]-1,3-thiazol-5-yl}ethan-1-one (2i). Colorless solid, yield = 72% (0.414 g), m.p. = 225–226 °C. IR (KBr) $v$ (cm⁻¹): 3443 (N-H), 1652 (C=O), 1588 (C=C), 668 (C-S-C). ¹H NMR (400 MHz, DMSO-$d_6$) $\delta$ (ppm): 2.30 (s, 3H, CH₃), 2.32 (s, 3H, CH₃), 2.40 (s, 3H, N=C-CH₃), 2.50 (s, 3H, COCH₃), 7.23 (d, 2H, $J$ = 8.4 Hz, Ar H), 7.69 (d, 2H, $J$ = 8.0 Hz, Ar H). HRMS [M + H] + calculated for $C_{15}H_{18}N_3OS$: ($m/z$) 288.1171, found 288.1161.

1-{4-Methyl-2-[N-2-[1-(4-methoxyphenyl)ethylidene]hydrazin-1-yl]-1,3-thiazol-5-yl} ethan-1-one (2j). Colorless solid, yield = 48% (0.291 g), m.p. = 198–200 °C. IR (KBr) $v$ (cm⁻¹): 3384 (N-H), 1653 (C=O), 1617 (C=N), 1595 (C=C), 668 (C-S-C). ¹H NMR (400 MHz, DMSO-$d_6$) $\delta$ (ppm): 2.18 (s, 3H, CH₃), 2.19 (s, 3H, CH₃), 2.37 (s, 3H, COCH₃), 3.98 (s, 3H, OCH₃), 6.99 (d, 2H, $J$ = 8.0 Hz, Ar H), 7.87 (d, 2H, $J$ = 8.0 Hz, Ar H). HRMS [M + H] + calculated for $C_{15}H_{18}N_3O_2S$: ($m/z$) 304.1120, found 304.1122.

1-{4-Methyl-2-[N-2-[1-(4-aminophenyl)ethylidene]hydrazin-1-yl]-1,3-thiazol-5-yl}ethan-1-one (2k). Brown solid, yield = 80% (0.462 g), m.p. = 222–223 °C. IR (KBr) $v$ (cm$^{-1}$): 3423 (N-H), 1699 (C=O), 1622 (C=N), 1558 (C=C), 654 (C-S-C). $^1$H NMR (400 MHz, DMSO) $\delta$ (ppm): 2.31 (s, 3H, CH$_3$), 2.40 (s, 3H, N=C-CH$_3$), 2.49 (s, 3H, COCH$_3$), 7.22 (d, 2H, $J$ = 8.8 Hz, Ar H), 7.81 (d, 2H, $J$ = 8.8 Hz, Ar H). HRMS [M + Na] + calculated for C$_{14}$H$_{16}$N$_4$NaOS: (m/z) 311.0943, found 311.0922.

1-{4-Methyl-2-[N-2-[1-(4-nitrophenyl)ethylidene]hydrazin-1-yl]-1,3-thiazol-5-yl}ethan-1-one (2l). Colorless solid, yield = 72% (0.459 g), m.p. = 245–246 °C. IR (KBr) $v$ (cm$^{-1}$): 3451 (N-H), 1690 (C=O), 1582 (C=C), 668 (C-S-C). $^1$H NMR (400 MHz, CDCl$_3$) $\delta$ (ppm): 2.46 (s, 3H, CH$_3$), 2.52 (s, 3H, COCH$_3$), 2.66 (s, 3H, N=C-CH$_3$), 7.95 (d, 2H, $J$ = 9.2 Hz, Ar H), 8.27 (d, 2H, $J$ = 8.8 Hz, Ar H). HRMS [M + H] $^+$ calculated for C$_{14}$H$_{15}$N$_4$O$_3$S: (m/z) 319.0865, found 319.0877.

## General procedure for the synthesis of 2-{N-2-[(furan)methylidene] hydrazin-1-yl}-1,3-thiazoles 2(m-r)

Thiosemicarbazide (0.005 mole) and substituted furaldehyde (0.005 mole) mixture was allowed to reflux at 80 °C in 25.0 mL of ethanol with continuous stirring until its completion. After the reaction completion, the mixture was cooled to room temperature and filtered to collect crude products. The crude products were recrystallized from ethanol to get substituted furaldehyde thiosemicarbazones 1(k-n) (Scheme 4). In the final step, substituted furaldehyde thiosemicarbazones (0.002 mole) and chlorocarbonyl derivatives (0.002 mole) mixture were refluxed (60 °C) in 20.0 mL of acetone with continuous stirring until its completion. The progress of the reactions was checked using the TLC method. Like the previous step, the reaction mixture was cooled to room temperature. The crude solid was separated by filtration and recrystallized from ethanol to yield final products 2(m–r) (Schemes 4–6).

2-[N-2-(Furan-2-ylmethylidene)hydrazine-1-yl]-4,5-dimethyl-1,3-thiazole (2m). Brown solid, yield = 50% (0.222 g), m.p. = 116–117 °C. IR (KBr) $v$ (cm$^{-1}$): 3378 (NH), 1635 (C=N),

**Scheme 4. Synthesis of furan-thiazole Schiff base hybrids 2(m–n). The Schiff base and thiazole skeletons are shown in blue and red lines, respectively.**

**Scheme 5. Synthesis of furan-thiazole Schiff base hybrids 2o. The Schiff base and thiazole skeletons are shown in blue and red lines, respectively.**

**Scheme 6. Synthesis of furan-thiazole Schiff base hybrids 2(p–r). The Schiff base and thiazole skeletons are shown in blue and red lines, respectively.**

1559 (C=C), 1106 (C-O), 769 (C-S-C). $^1$H NMR (400 MHz, DMSO-$d_6$) δ (ppm): 2.12 (s, 3H, CH$_3$), 2.16 (s, 3H, CH$_3$), 6.63 (d, 1H, $J$ = 1.6 Hz, furan), 6.96 (d, 1H, $J$ = 3.2 Hz, furan), 7.85 (s, 1H, furan), 8.21 (s, 1H, CH=N). HRMS [M]$^+$ calculated for C$_{10}$H$_{11}$N$_3$OS: ($m/z$) 221.0623, found 221.0617.

2-[N-2-(4-Methylfuran-2-ylmethylidene)hydrazine-1-yl]-4,5-dimethyl-1,3-thiazole (2n). Brown solid, yield = 58% (0.273 g), m.p. = 192–193 °C. IR (KBr) $v$ (cm$^{-1}$): 3447 (NH), 1577 (C=C), 1205 (C-O), 780 (C-S-C). $^1$H NMR (400 MHz, DMSO-$d_6$) δ (ppm): 2.14 (s, 3H, CH$_3$), 2.19 (s, 3H, CH$_3$), 2.35 (s, 3H, CH$_3$), 6.29 (dd, 1H, $J$ = 2.4 Hz, 1.6 Hz, furan), 6.88 (d, 1H, $J$ = 3.2 Hz, furan), 8.14 (s, 1H, CH=N). HRMS [M+H]+ calculated for C$_{11}$H$_{14}$N$_3$OS: ($m/z$) 236.0858, found 236.0859.

Ethyl-2-[N-2-(4-methylfuran-2-ylmethylidene)hydrazine-1-yl]-1,3-thiazole-4-acetate (2o). Green solid, yield = 60% (0.352 g), m.p. = 108–109 °C. IR (KBr): $v$ (cm$^{-1}$): 3388 (NH), 1731 (C=O), 1625 (C=N), 1537 (C=C), 1184 (C-O), 791 (C-S-C). $^1$H NMR (400 MHz, DMSO-$d_6$) δ (ppm): 1.21 (t, 3H, $J$ = 7.8 Hz, CH$_2$-CH$_3$), 2.34 (s, 3H, CH$_3$), 3.67 (s, 2H, CH$_2$-CO), 4.03 (q, 2H, $J$ = 7.2 Hz, CH$_2$-CH$_3$), 6.26 (dd, 1H, $J$ = 3.6 Hz, 1.2 Hz, furan), 6.74 (s, 1H, =CH-S), 6.79 (d, 1H, $J$ = 2.8 Hz, furan), 7.99 (s, 1H, CH=N). HRMS [M]+ calculated for C$_{13}$H$_{15}$N$_3$O$_3$S: ($m/z$) 293.0834, found 293.0835.

1-{2-[N-2-(Furan-2-ylmethylidene)hydrazin-1-yl]-4-methyl-1,3-thiazol-5-yl}ethan-1-one (2p). Yellow solid, yield = 50% (0.249 g), m.p. = 165–166 °C. IR (KBr) $v$ (cm$^{-1}$): 3410 (NH), 1656 (C=O), 1639 (C=N), 1557 (C=C), 1157 (C-O), 777 (C-S-C). $^1$H NMR (400 MHz, DMSO-$d_6$) δ (ppm): 2.39 (s, 3H, CH$_3$), 2.50 (s, 3H, COCH$_3$), 6.63 (d, 1H, $J$ = 1.2 Hz, furan), 6.89 (d, 1H, $J$ = 2.8 Hz, furan), 7.84 (s, 1H, furan), 8.03 (s, 1H, CH=N). HRMS [M+H]+ calculated for C$_{11}$H$_{12}$N$_3$O$_2$S: ($m/z$) 250.0650, found 250.0665.

1-{2-[N-2-(5-Bromofuran-2-ylmethylidene)hydrazin-1-yl]-4-methyl-1,3-thiazol-5-yl}ethan-1-one (2q). Yellow solid, yield = 92% (0.604 g), m.p. = 145–146 °C. IR (KBr) $v$ (cm$^{-1}$): 3420 (NH), 1642 (C=O), 1622 (C=N), 1564 (C=C), 1133 (C-O), 731 (C-S-C). $^1$H NMR (400 MHz, DMSO-$d_6$) δ (ppm): 2.41 (s, 3H, CH$_3$), 2.51 (s, 3H, COCH$_3$), 6.76 (d, 1H, $J$ = 3.6 Hz,

furan), 6.94 (d, 1H, $J$ = 3.2 Hz, furan), 7.97 (s, 1H, CH=N). HRMS [M+H] + calculated for $C_{11}H_{11}BrN_3O_2S$: (*m/z*) 327.9755, found 327.9770.

1-{2-[N-2-(5-Nitrofuran-2-ylmethylidene)hydrazin-1-yl]-4-methyl-1,3-thiazol-5-yl} ethan-1-one (2r). Orange solid, yield = 70% (0.412 g), m.p. = 220–221°C. IR (KBr) $v$ (cm$^{-1}$): 3412 (NH), 1676 (C=O), 1623 (C=N), 1583 (C=C), 1199 (C-O), 736 (C-S-C). $^1$H NMR (400 MHz, DMSO-$d_6$) $\delta$ (ppm): 2.42 (s, 3H, CH$_3$), 2.50 (s, 3H, COCH$_3$), 7.18 (d, 1H, $J$ = 4.0 Hz, furan), 7.77 (d, 1H, $J$ = 4.0 Hz, furan), 8.08 (s, 1H, CH=N). HRMS [M+H]$^+$ calculated for $C_{11}H_{11}N_4O_4S$: (*m/z*) 295.0501, found 295.0488.

## Antimicrobial activity assay

According to a previous report, the agar disk diffusion technique was used to examine the *in vitro* antimicrobial activities of the synthesized compounds [36]. Basal media for bacterial strains were prepared using Mueller Hinton Agar (HIMEDIA, India). Prepared media were incubated for 24 h and scanned for contamination because only non-contaminated dishes were chosen for this assay. The inoculation process of the test organism on media was performed by a sterile cotton bar. The disks containing the sample were smoothly placed on pre-inoculated agar plates. The plates were then incubated aerobically at 37 °C for 24 h for the antibacterial assays. This experiment used DMSO (dimethyl sulfoxide) as a control, and ceftriaxone was applied as a standard. Each disk was charged with 25 µL of sample solution in DMSO, which contained 300 µg of synthesized compounds. On the other hand, each disk was charged with 10 µL of ceftriaxone solution in DMSO, which includes 50 µg of standard ceftriaxone. The dishes were then incubated for 24 h, and the diameter of inhibition zones (mm) was measured using a measuring scale. In this study, our experiment was operated on two gram-positive bacteria, namely, *Staphylococcus aureus* (cars-2) and *Bacillus subtilis* (carsgp-3), and two gram-negative bacteria such *Escherichia coli* (carsgn-2) and *Salmonella typhimurium* (JCM-1652).

## *In silico* studies

**Molecular docking studies.** Molecular docking studies were performed to investigate the binding mode of synthesized thiazole analogs with the selected proteins based on maximum binding affinity and minimal binding energy with respective proteins. The target crystal structure of proteins (a) *B. subtilis* (PDB ID: 6JHK) [37] and (b) E. coli (PDB ID: 1KZN) [38] was collected from the protein data bank (https://www.rcsb.org). The target proteins were prepared using PyMol (version 2.4) after eliminating all bound water, heteroatoms, and co-crystallized ligands. Energy minimization of the selected proteins was completed by Swiss-PdbViewer. Structure optimization was executed *via* Gaussian 09 software based on B3LYP/6-31+G (d, p) basis set up in the DFT method. The DFT method is a versatile tool for delving deeper into the binding affinity and ligand interactions with receptors. This method also provides spectroscopic data, dipole moment, electronic structure, and energy calculation, which are crucial to characterize the structure of synthesized compounds, understanding the reaction mechanism, and overall drug design [39–41]. AutoDock Vina in PyRx 0.8 software was implemented for docking studies maintaining standard protocol, and Discovery Studio 4.2 was utilized to envisage the final geometries of resulting complexes. The binding mode with the highest negative binding energy score was selected for the ultimate presentation.

**ADMET prediction.** Molinspiration tool of online version was applied to determine the ADME (absorption, distribution, metabolism, and excretion) properties (https://www. molinspiration.com). The absorption percent (%ABS) was calculated using the following formula (Eq. 2).

$$\%ABS = 109 - \left(0.3459 \times TPSA\right) \qquad (2)$$

Here, the topological polar surface area is abbreviated as TPSA.

The BOILED-Egg model was generated using the online version of the SwissADME tool. Four toxicity parameters: (a) mutagenic, (b) tumorigenic, (c) irritant, and (d) reproductive effects with drug-likeness and drug-score properties were evaluated by Osiris Property Explorer (https://www.organic-chemistry.org/prog/peo) according to a previously described report [42].

**Molecular dynamics.** Molecular dynamics (MD) simulations were conducted on two complexes, each comprising a protein bound to a compound having a maximum docking score. Prior to the MD simulation, these complexes were positioned within a cubic periodic box and solvated with simple point charge (SPC) water molecules during the topology process. To achieve each system neutralization, varying numbers of sodium ions were introduced. A 50,000-step steepest descent minimization was applied for energy minimization. Following this, a 100 ps NVT (constant Number of particles, Volume, and Temperature) simulation was performed, succeeded by a 100 ps NPT (constant Number of particles, Pressure, and Temperature) ensemble at 300 K and 1.0 bar, employing a Berendsen thermostat and barostat (ref) to equilibrate water molecules surrounding the complexes. The particle mesh Ewald (PME) algorithm [43] was utilized to account for long-range Coulomb interactions, and Lennard-Jones potentials were applied for van der Waals interactions, with a cutoff of 1.2 nm considering both long- and short-range interactions.

The LINCS algorithm was utilized to constrain all bonds [44]. Subsequently, a 100 ns long MD simulation was conducted in the NVT ensemble at 300 K, employing a time step of 2.0 fs and saving final coordinates at 10.0-ps intervals. The MD simulation was performed using GROMACS 5.1.4 [45] and the Charmm36 force field [46].

**MM-PBSA binding energy calculation.** The Molecular Mechanics Poisson-Boltzmann Surface Area (MM-PBSA) method serves as a valuable tool for assessing the binding affinity in biomolecular systems by calculating the interaction free energy. The binding energy was determined using MM-PBSA protocols implemented in the g_mmpbsa package with GROMACS MD trajectories. The calculation of binding energies is based on the following equation:

$$\Delta G_{BE} = G_C + \left(G_P + G_L\right) \qquad (3)$$

where $\Delta G_{BE}$ is the binding free energy between a protein and ligand. $G_C$, $G_P$, and $G_L$ are the total free energies of the complex (protein-ligand), protein and ligand, respectively. The free energies of each individual entity can be expressed as:

$$G_X = E_{MM} + G_{solv} - TS \qquad (4)$$

where $X$ denotes the complex, protein or ligand. Here, $E_{MM}$ represents the energy of molecular mechanics (MM) potential, incorporating bonding and non-bonding terms.

$$E_{MM} = E_{bonded} + E_{non-bonded} = E_{bonded} + \left(E_{vdW} + E_{elec}\right) \qquad (5)$$

where $E_{bonded}$ is bonded interaction consisting of bond, angle, dihedral and improper interactions. The $E_{bonded}$ includes both van der Waals and electrostatic interactions. The free energy of solvation is defined by the energy required to transfer a solute from vacuum into the solvent. In MM-PBSA method, it is computed in an implicit solvent model. The solvation free energy ($G_{solv}$) is defined as:

$$G_{solv} = G_{polar} + G_{non-polar} \qquad (6)$$

where $G_{polar}$ and $G_{non-polar}$ are the electrostatic and non-electrostatic contributions to the solvation free energy, respectively.

## Results and discussion

### Chemistry

In the present study, eighteen thiazole Schiff base derivatives were synthesized with substituted phenyl and furan ring with substitution, and also by changing the substituents of the thiazole ring. In the first instance, several phenyl rings containing thiazole Schiff base analogs **2(a–l)** were synthesized. Intermediate thiosemicarbazones with substituted phenyl ring **1(a–j)** were prepared by the reaction between thiosemicarbazides and substituted benzaldehydes or acetophenones (Scheme 1). Target compounds **2(a–l)** were synthesized by the reaction between thiosemicarbazones and chlorocarbonyl derivatives with 48–92% yields (Schemes 2 and 3). Compounds (**2a**) and (**2i**) were recently synthesized to evaluate their anticancer activity [18,47]. Subsequently, furan-thiazole Schiff base hybrids **2(m–r)** were synthesized. Thiosemicarbazone intermediates with substituted furan ring **1(k–n)** was prepared by the reaction between thiosemicarbazides and substituted for furaldehydes (Scheme 4). Target compounds **2(m–r)** were prepared in 50–92% yields by the reaction between thiosemicarbazones **1(k–n)** and chlorocarbonyl derivatives (Schemes 4–6). Compound (**2p**) was reported to have significant antituberculosis activity against *Mtb*, $H_{37}Rv$. At the same time, this compound was described as an antimalarial agent [48,49]. The synthetic strategy of all the synthesized compounds is summarized in **Fig 2**.

All the synthesized Schiff base analogs have been characterized using IR, [1]H NMR, and mass (HRMS) spectroscopic techniques. For (**2a**), the peaks at 3455, 3100, 1626, 1497, 1113, and 760 cm$^{-1}$ in the IR spectrum were assigned for NH, OH, C=N, C=C of Ar, C–O, and C–S–C groups, respectively. The [1]H NMR spectrum of (**2a**) demonstrated two singlets for three protons at 2.24 and 2.26 ppm for methyl groups at thiazole C–4 and thiazole C–5, respectively. Aromatic protons were observed in between 6.94 to 7.34 ppm. A singlet of one proton for the existence of –CH=N– appeared at 8.52 ppm. Calculated mass (HRMS) data agreed with the experimental values, which confirmed the structures of the respective compounds.

### Antibacterial activity

The i*n vitro* antibacterial activities of the synthesized derivatives, taking two gram-positive bacteria and two gram-negative bacteria, were determined using the agar disk diffusion technique. **Table 1** displayed the zone of inhibition produced by the standards and synthesized derivatives.

In the first series, compounds **2(a-d)** were moderately active against most bacterial strains. The azomethine group with methyl substituents revealed comparatively higher activity, like (**2c**) and (**2d**), whereas the para-substituted fluoro group in (**2d**) exhibited maximum against *B. Subtilis* (48.3 ± 0.6 mm). In the second series, (**2e**), (**2g**), and (**2h**) showed no activity, which might be the presence of bromo and dichloro substituents in the phenyl ring. However, para-substituted fluoro group-containing derivative (**2f**) showed some moderate activity against three bacterial strains. The para position replaced with methyl and methoxy group in (**2i**) and (**2j**) had reduced their activity against two strains. The introduction of the amine group into the 3-position of the phenyl ring in (**2k**) revealed moderate activity, while replacement with the nitro group in (**2l**) showed improved inhibition value against *B. subtilis*.

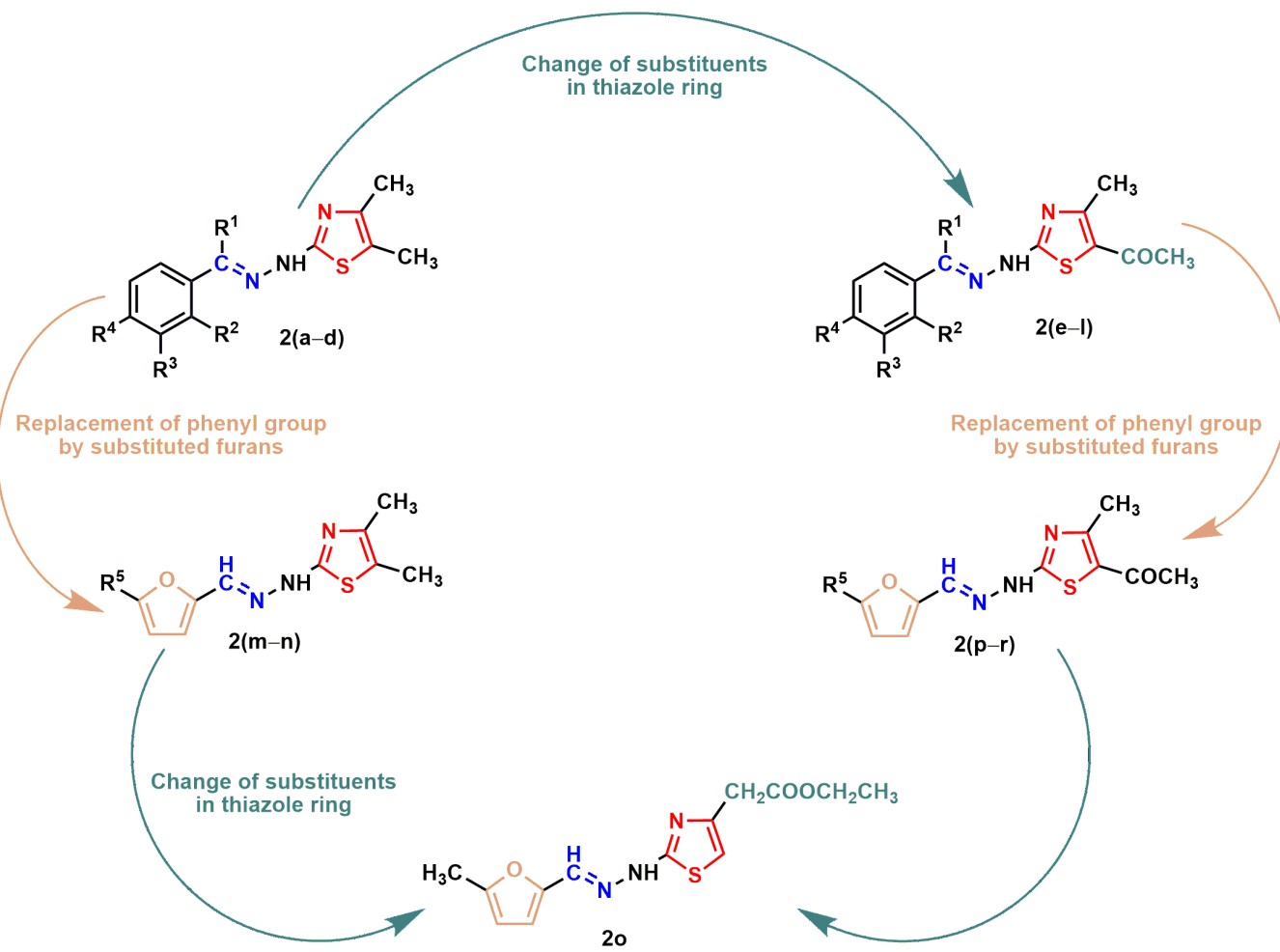

**Fig 2. Summary for the synthesis of aryl and heteroaryl ring containing thiazole Schiff bases reported in this study.**

In case of the furan ring containing derivatives, methyl substituted thiazole ring and the furan ring in (**2m**) showed good activity against all strains. However, replacing the methyl group in the 5-position of the furan ring in (**2n**) exhibited excellent activity against *B. subtilis* with an inhibition zone of 45.3 ± 0.6 mm. In compound (**2o**), a new substituent was introduced in the thiazole ring, dropping the activity from **2(m-n)**. In the last three derivatives, the thiazole ring was replaced with methyl and methoxy groups in the 4 and 5 positions for (**2p**), (**2q**), and (**2r**). Moreover, the 5-position of the furan ring of these derivatives was replaced by bromo and nitro groups in (**2q**) and (**2r**), respectively, while (**2p**) had no substituent. Among these three derivatives, the nitro group containing (**2r**) had higher activity than (**2p**) and (**2q**) against three strains, where they showed no activity against *E. coli*. As compounds **2d** and **2n** showed the maximum activity against *B. subtilis*, the minimal inhibitory concentrations (MICs) of compounds **2d** and **2n** against *B. subtilis* were determined. The MICs values in both cases were 1 µg/mL.

### *In silico* studies

**Molecular docking studies.** *In silico* molecular docking was performed using Gaussian 09, PyRx 0.8, and Pymol software packages to realize the mode of action of synthesized

**Table 1. Zone of inhibition diameters (mm) of the synthesized derivatives 2(a–r) and standard ceftriaxone (Cef) against gram-positive and gram-negative bacterial strains.**

| Compd. | Gram (+) bacteria | | Gram (–) bacteria | |
|---|---|---|---|---|
| | *S. aureus* | *B. subtilis* | *E. coli* | *S. typhimurium* |
| 2a | 10.7 ± 0.6 | 27.3 ± 0.6 | 9.0 ± 1.0 | 20.7 ± 0.6 |
| 2b | 10.0 ± 1.0 | 17.0 ± 1.0 | 13.3 ± 1.5 | 12.0 ± 2.0 |
| 2c | 22.7 ± 0.6 | 28.3 ± 0.6 | – | 21.0 ± 1.0 |
| 2d | 18.3 ± 1.5 | 48.3 ± 0.6 | 16.3 ± 0.6 | 22.7 ± 0.6 |
| 2e | – | – | – | – |
| 2f | 14.0 ± 1.0 | – | 15.7 ± 0.6 | 17.3 ± 1.5 |
| 2g | – | – | – | – |
| 2h | – | – | – | – |
| 2i | 17.7 ± 0.6 | – | – | 13.3 ± 1.5 |
| 2j | – | – | 15.0 ± 1.0 | – |
| 2k | 17.3 ± 1.5 | 14.7 ± 1.5 | – | 11.0 ± 2.0 |
| 2l | – | 26.0 ± 1.0 | – | – |
| 2m | 13.3 ± 1.5 | 18.3 ± 1.5 | 15.3 ± 1.5 | 17.3 ± 0.6 |
| 2n | 18.3 ± 0.6 | 45.3 ± 0.6 | 23.3 ± 0.6 | 9.3 ± 2.5 |
| 2o | 14.0 ± 2.0 | 32.0 ± 1.0 | 9.0 ± 1.0 | 18.7 ± 0.6 |
| 2p | 11.3 ± 2.5 | 9.0 ± 2.0 | – | 9.0 ± 2.0 |
| 2q | 8.0 ± 2.0 | 9.3 ± 2.5 | – | 8.7 ± 1.5 |
| 2r | 25.3 ± 0.6 | 27.0 ± 1.0 | – | – |
| Cef | 40.3 ± 0.6 | 20.0 ± 1.0 | 38.3 ± 0.6 | 44.3 ± 0.6 |
| DMSO | – | – | – | – |

[1]The data are mean ± SD (standard deviation) (n = 3).

– Represents no activity.

compounds against target receptors. Structure optimization of the selected compounds was accomplished by the DFT method in Gaussian 09 software based on the B3LYP/6-31 + G (d, p) level of theory, as illustrated in **Fig 3**.

Compounds (**2d**) and (**2n**) were docked against co-crystals of *B. subtilis* (PDB ID: 6JHK) and *E. coli* (PDB ID: 1KZN), respectively, considering the docking score and how closely the ligands were bound theoretically to the active site of the receptor proteins. Initially, we docked the compounds on the basis of their wet lab activity with various protein receptors, then verified the active site of the receptor proteins from the obtained structure and compared it with the interaction area of our selected ligands (**2d**) and (**2n**) for final presentations [50–52]. **Fig 4** presents the 2D and 3D non-covalent interactions between the selected compounds and target receptors.

The poses with the highest negative docking scores were selected for the final presentation that signifies the best affinities of the compounds towards the target receptors. The number of hydrogen bonds between the docked compound and the amino acid residues of the receptor protein plays a crucial role in displaying the bioactivity of a drug molecule. Compound (**2d**) showed a binding score of -6.84 Kcal/mol when docked against 6JHK. The electronegativity of nitrogen and sulfur enhances the interaction with biomolecules, and phenyl ring presence in compound (**2d**) played a crucial role in hydrophobic interaction with the receptor. While docking, the hydrazinyl N-H of this compound exhibited hydrogen bond interaction with ILE102 at a distance of 1.79 Å. The nitrogen on thiazole ring displayed hydrophobic interaction with LYS73 at a distance of 4.89 Å. The thiazole methyl was engaged in a hydrophobic

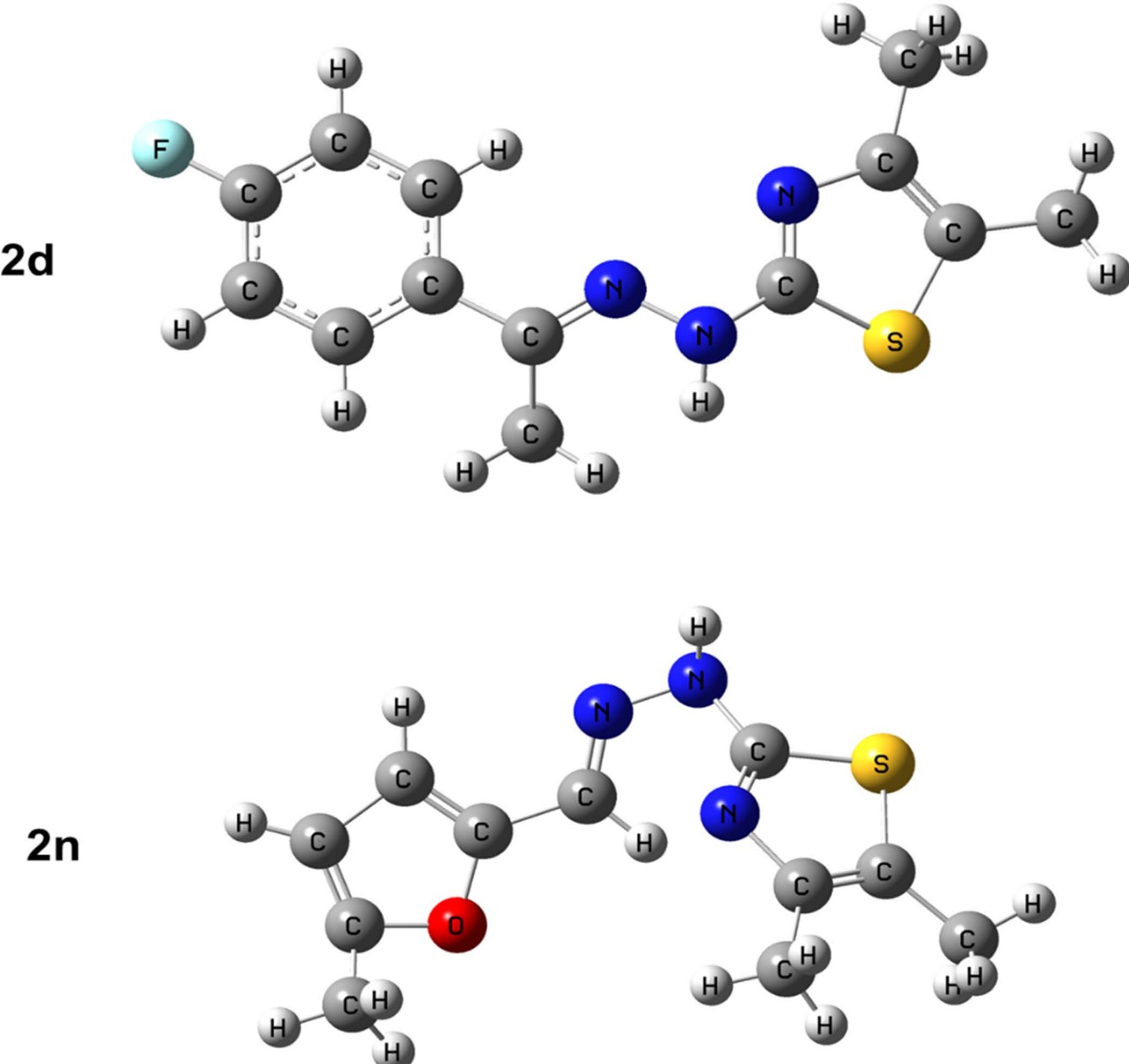

**Fig 3. Optimized molecular structures of the synthesized derivatives (2d) and (2n). Structure optimization was carried out applying DFT/ B3LYP/6-31 + G(d,p) basis setup.**

pi-alkyl interaction with LEU74 and ALA 98 at distances of 4.49 Å and 3.66 Å, respectively. On the other hand, the phenyl group was involved in pi-alkyl interactions with LYS78 and LYS73 at distances of 5.02 Å and 3.91 Å, respectively. Moreover, the phenyl methyl group showed two halogen interactions with GLY76 and ALA77 at distances of 3.08 Å and 3.63 Å, respectively. The thiazole ring was also associated with hydrophobic pi-alkyl interaction with LEU74 at a distance of 5.47 Å. When docked against 1KZN protein, compound (**2n**) exhibited the lowest binding affinity with the amino acid residues with a binding score of -6.47 kcal/

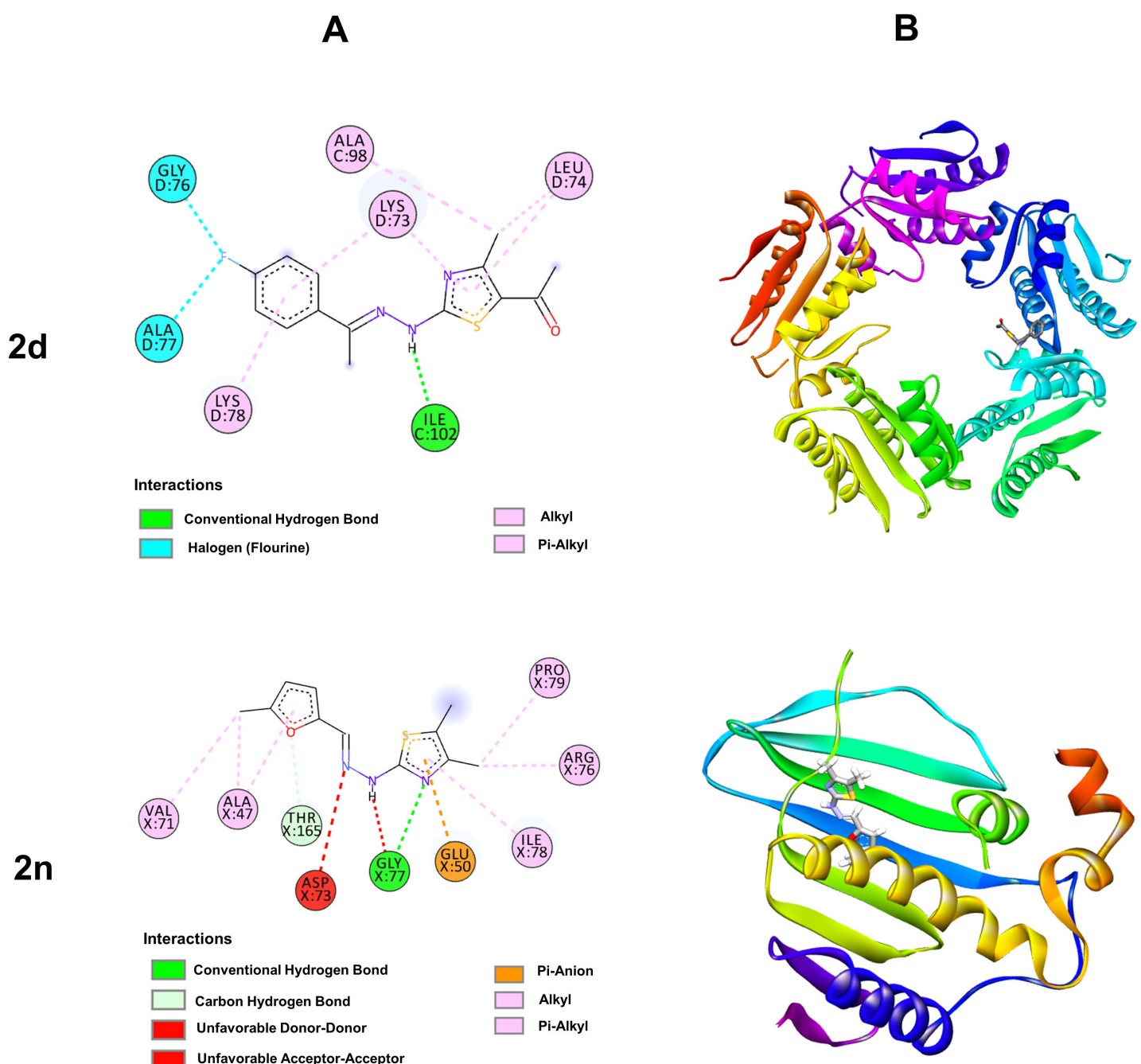

**Fig 4. Molecular docking studies of (2d) and (2n) against 6JHK and 1KZN protein receptors, respectively. (A) 2D interaction sketches. (B) 3D docking predictions.**

mol. The nitrogen in thiazole and π-electron of the furan ring were engaged in hydrogen bonding interactions with GLY77 and THR165 at distances of 2.88 Å and 2.45 Å, respectively. Electrostatic pi-anion interaction was found between the π-electron of the thiazole ring and GLU50 at a distance of 3.71Å. The π-electron and methyl in thiazole were also engaged in hydrophobic interactions with PRO79, ARG76, and ILE78 with distances of 4.46 Å, 4.40 Å and 4.76 Å, respectively. On the other hand, furan ring has the capability to weaken the cell

membrane and created a strong interaction with various protein receptor [53]. Besides, the substituent methyl group of the furan ring exhibited hydrophobic interaction with ALA47 and VAL71 at distances of 3.57 Å and 4.08 Å, respectively.

*In silico* **ADMET prediction.** Many new drugs fail in clinical trials due to unfavorable ADME and toxicity risk. Therefore, it is significant to know the pharmacokinetic properties during the drug design and development process. Two well-known rules, namely (1) Lipinski's rule of five and (2) Veber's rule, proved useful in predicting the oral bioavailability of drug molecules. In the context of drug discovery, Lipinski's rule speculates that bad permeation or absorption is more possibly when there are more than 10 H-bond acceptors (HBA), 5 H-bond donors (HBD), lipophilicity (cLogP) is greater than 5, and the molecular weight (MW) is greater than 500. On the other hand, Veber's rule predicts that oral bioavailability is preferable if the molecule maintains number of rotatable bonds (NROTB) and Topological polar surface area (TPSA) is equal to or less than 10 and 140 Å$^2$ [54,55]. The ADMET (adsorption, distribution, metabolism, excretion, and toxicity) properties are significantly influenced by the lipophilicity and TPSA parameters. Molecules having higher lipophilicity (clogP'5) exhibit poor adsorption (%ABS) and solubility (logS). In contrast, low lipophilicity showed poor ADMET properties. All our synthesized compounds showed lipophilicity values just below 5 which justify oral bioavailability. The percentage of the absorbance (%ABS) was more than 80% except for (**2l**) and (**2r**), inferring oral bioavailability to the systemic circulation in an active form. TPSA value is correlated with H-bonding and is one of the important indicators for drug oral bioavailability. All the synthesized compounds described in this paper represent the TPSA values within the acceptable limit (57.51–113.32) (**Table 2**).

**Table 2. Speculated pharmacokinetic properties of synthesized derivatives 2(a–r) and ceftriaxone (Cef).**

| Compd. | Lipinski's Violations | Lipinski's rule | | | | Veber's rule | | logS | %ABS |
|---|---|---|---|---|---|---|---|---|---|
| | | MW (≤500) | HBA (≤10) | HBD (≤5) | clogP (≤5) | NROTB (≤10) | TPSA (140 Å²) | | |
| 2a | 0 | 247.32 | 4 | 2 | 2.67 | 3 | 57.51 | −3.41 | 89.16 |
| 2b | 0 | 276.32 | 6 | 1 | 2.66 | 4 | 83.11 | −4.16 | 80.33 |
| 2c | 0 | 324.25 | 3 | 1 | 3.96 | 3 | 37.28 | −4.90 | 96.14 |
| 2d | 0 | 263.34 | 3 | 1 | 3.34 | 3 | 37.28 | −4.38 | 96.14 |
| 2e | 0 | 352.26 | 4 | 1 | 3.88 | 4 | 54.35 | −5.24 | 90.25 |
| 2f | 0 | 291.35 | 4 | 1 | 3.25 | 4 | 54.35 | −4.72 | 90.25 |
| 2g | 0 | 352.26 | 4 | 1 | 3.90 | 4 | 54.35 | −5.24 | 90.25 |
| 2h | 0 | 342.25 | 4 | 1 | 4.37 | 4 | 54.35 | −5.88 | 90.25 |
| 2i | 0 | 287.39 | 4 | 1 | 3.54 | 4 | 54.35 | −4.75 | 90.25 |
| 2j | 0 | 303.39 | 5 | 1 | 3.15 | 5 | 63.59 | −4.42 | 87.06 |
| 2k | 0 | 288.38 | 5 | 3 | 2.17 | 4 | 80.38 | −4.48 | 81.27 |
| 2l | 0 | 318.36 | 7 | 1 | 3.05 | 5 | 100.18 | −4.86 | 74.44 |
| 2m | 0 | 221.28 | 4 | 1 | 1.99 | 3 | 50.42 | −3.39 | 91.60 |
| 2n | 0 | 235.31 | 4 | 1 | 2.21 | 3 | 50.42 | −3.75 | 91.60 |
| 2o | 0 | 293.35 | 6 | 1 | 2.42 | 7 | 76.73 | −3.36 | 82.53 |
| 2p | 0 | 249.29 | 5 | 1 | 1.90 | 4 | 67.49 | −3.73 | 85.72 |
| 2q | 0 | 328.19 | 5 | 1 | 2.83 | 4 | 67.49 | −4.32 | 85.72 |
| 2r | 0 | 294.29 | 8 | 1 | 1.98 | 5 | 113.32 | −4.71 | 69.90 |
| Cef | 1 | 554.0 | 14 | 4 | -2.95 | 8 | 287.3 | −2.95 | 9.62 |

The boiled egg model is rapid and the most accurate predictive method for the potency evaluation of a drug candidate in the drug discovery and development process. This model functions based on the calculation of lipophilicity and polarity of the molecules [56,57]. Generally, this model is used to estimate the pharmacokinetic properties such as human intestinal adsorption (HIA) and blood-brain barrier (BBB) values. According to ADMET, compounds having HIA values greater than 80% indicate good permeation across the membrane and can easily reach in hepatic portal vein [58]. Smoothly cross the blood brain barrier, a compound must have a positive BBB value. In contrast, the negative values signify poor distribution in brain. The predicted HIA and BBB values of the synthesized compounds are illustrated in **Fig 5**.

All the synthesized compounds **2(a–r)** displayed HIA values greater than 80%. The compounds showed positive BBB values as well. The summary of the adsorption properties of synthesized compounds are displayed in **Fig 6**.

The white region indicates the physicochemical space of compounds with the highest possibility of absorption by the gastrointestinal tract. Whereas, the yellow region indicates the physicochemical space of compounds with the highest possibility of penetration to the brain [56]. Most of the synthesized compounds have fallen in the white region except (**2c**), (**2d**), and (**2n**) which predicted that the designed molecules could have better HIA properties. Whereas only compound (**2r**) has fallen outside of the yellow and white region.

The toxicity risk factor is a remarkable parameter in drug discovery, as 30% of drug molecules cannot pass the clinical trial due to their high toxicity risk [59]. Toxicity is usually generated due to the presence of a particular fragment of a drug molecule. This study evaluated toxicity parameters such as mutagenicity, tumorigenicity, irritancy, and reproductive effects of the synthesized compounds and standard ceftriaxone. The calculated toxicity values are summarized in **Table 3**.

The results revealed that all the synthesized compounds exhibited low irritant values, reproductive effects, and mutagenic risk factors, whereas (**2n**), (**2o**), (**2q**), and (**2r**) were predicted to show high mutagenic risk factors with exceptions. The predicted drug-likeness values of the synthesized compounds and standards are also presented in **Table 3**. The predicted drug-score of the synthesized compounds **2(a–r)** were ranging from 0.15–0.45, and the values are summarized in **Fig 7** together with standard ceftriaxone for comparison.

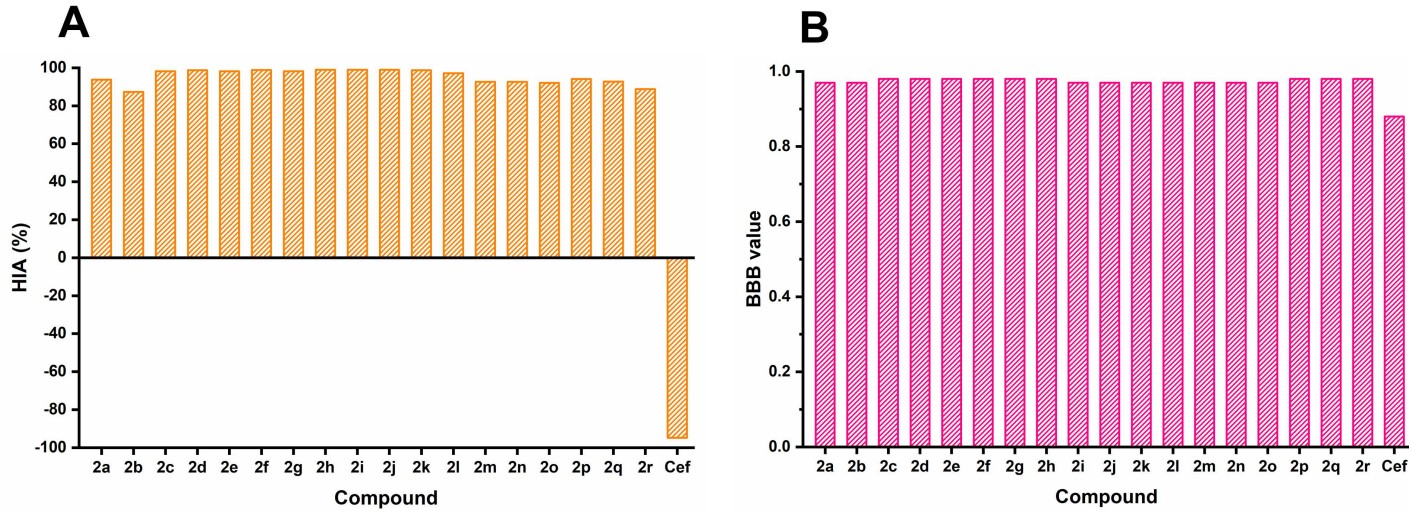

**Fig 5. Predicted HIA (%) and BBB values of synthesized derivatives 2(a–r) and ceftriaxone (Cef).**

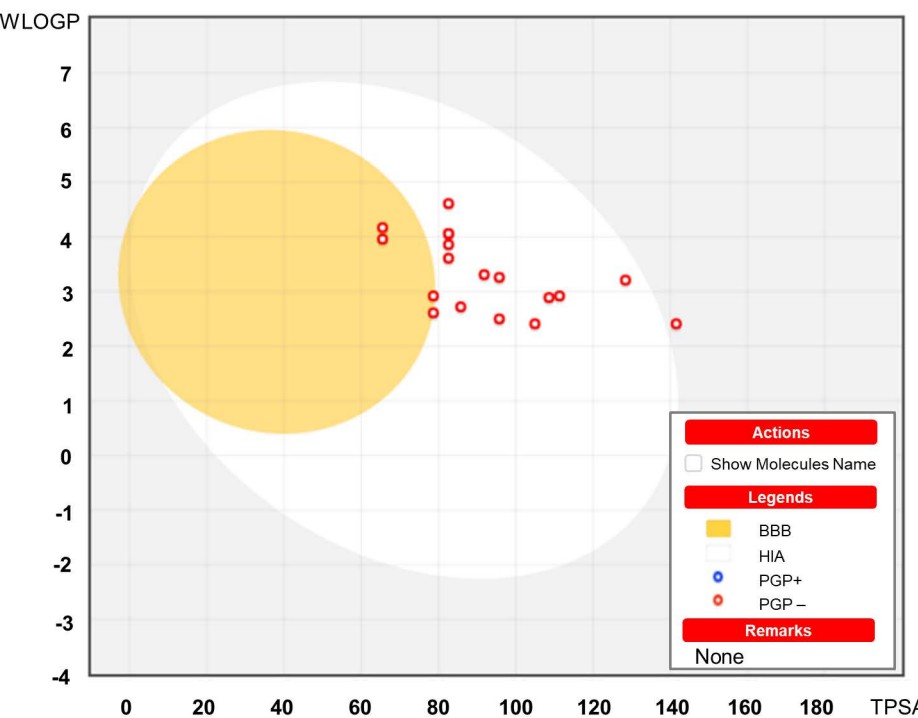

**Fig 6. Outline of the predictive BOILED-Egg model.** The white region indicates the physicochemical space of compounds with highest possibility of absorption by the gastrointestinal tract, and the yellow yolk region indicates the physicochemical space of compounds with the highest possibility of penetration to the brain.

**Table 3.** *In silico* toxicity effects and drug-likeness values of synthesized analogs 2(a–r) and ceftriaxone. The toxicity effects are shown as M (mutagenic), T (tumorigenic), I (irritant), and R (reproductive).

| Compd. | Toxicity effects | | | | Drug-Likeness |
|---|---|---|---|---|---|
| | M | T | I | R | |
| 2a | Low | High | Low | Low | 1.3 |
| 2b | Low | High | Low | Low | -3.86 |
| 2c | Low | High | Low | Low | -2.75 |
| 2d | Low | High | Low | Low | -0.08 |
| 2e | Low | High | Low | Low | -0.25 |
| 2f | Low | High | Low | Low | 2.35 |
| 2g | Low | High | Low | Low | 1.3 |
| 2h | Low | High | Low | Low | 4.17 |
| 2i | Low | High | Low | Low | 1.85 |
| 2j | Low | High | Low | Low | 3.47 |
| 2k | Low | High | Low | Low | 1.93 |
| 2l | Low | High | Low | Low | -6.99 |
| 2m | Low | High | Low | Low | 1.21 |
| 2n | High | High | Low | Low | 1.20 |
| 2o | High | High | Low | Low | -6.7 |
| 2p | Low | High | Low | Low | 3.74 |
| 2q | High | High | Low | Low | 2.42 |
| 2r | High | High | Low | Low | 2.19 |
| Cef | Low | Low | Low | Low | 16.69 |

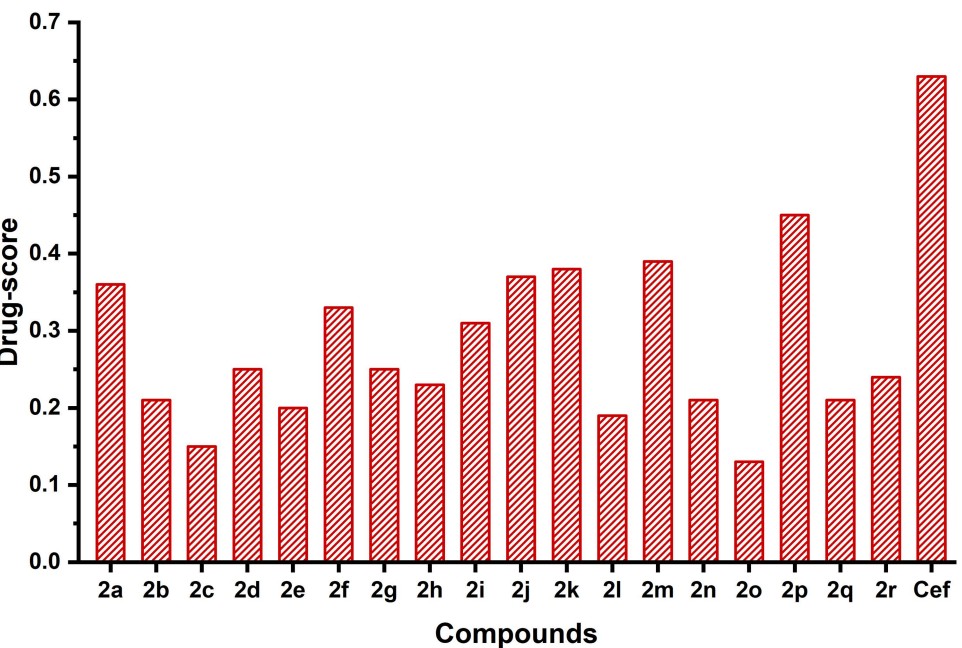

**Fig 7. Speculated drug-score values of the synthesized derivatives 2(a–r) and ceftriaxone (Cef).**

**MD simulation.** To study the binding stability of docked complexes in an aqueous solution, we initially assessed the root mean square displacement (RMSD) and The Root Mean Square Fluctuation (RMSF). The binding output trajectories of two different complexes were represented in **Fig 8**.

The detailed RMSD of ligands within complexes, relative to equilibrated initial structures, was analyzed and depicted in **Fig 9**. The ligand (**2d**) exhibited a variation in RMSD from 0.3-0.4 nm, while (**2n**) ranged from 0.15-0.30 nm during the 100 ns. From **Fig 9** and through visual inspection (**Fig 8**), it was evident that these two ligands (**2d**) and (**2n**) remained within the interacting sites of proteins during the whole simulation time and fluctuated in a minimal range. Next, we examined the contribution of individual protein residues to structural fluctuations and their intensity using RMSF (**Fig 10**).

The Root Mean Square Fluctuation (RMSF) functions as a valuable tool for evaluating the flexibility of protein residues. It reveals the extent to which fluctuations vary along the protein chain, both before and after binding with a ligand. In complexes with (**2d**) and (**2n**), protein residues exhibited minimal fluctuations within a range of 0.60 nm. Notably, within the five protein chains of 6JHK, the maximum fluctuations (0.59 and 0.54 nm) are observed in residues GLU118 and LEU119 in chain C. Despite this, the ligand (**2d**) binds in the active site of 6JHK, interacting with residues ALA98, LEU99, GLU100, ILE102, GLU103, and THR104 in chain C, as well as ASN45, THR70, LYS73, LEU74, GLY76, ALA77, and LYS78 in chain D. RMSF values for chain C residues ALA98, LEU99, GLU100, ILE102, GLU103, and THR104 range from 0.17 nm to 0.23 nm, and for chain D residues ASN45, THR70, LYS73, LEU74, GLY76, ALA77, and LYS78, they range from 0.13 nm to 0.19 nm. The lower fluctuations observed in these residues contribute to the effective binding of ligand (**2d**) within the active site of 6JHK.

Similarly, in protein 1KZN, residues GLY15, and LEU16 exhibit the maximum fluctuations of 0.61 nm and 0.60 nm, respectively. The ligand (**2n**) located in the active site of 1KZN, interacts with residues exhibiting lower fluctuations (0.06 ~ 0.12 nm), contributing to its effective

| Complex | Initial configuration | Final configuration |
| --- | --- | --- |

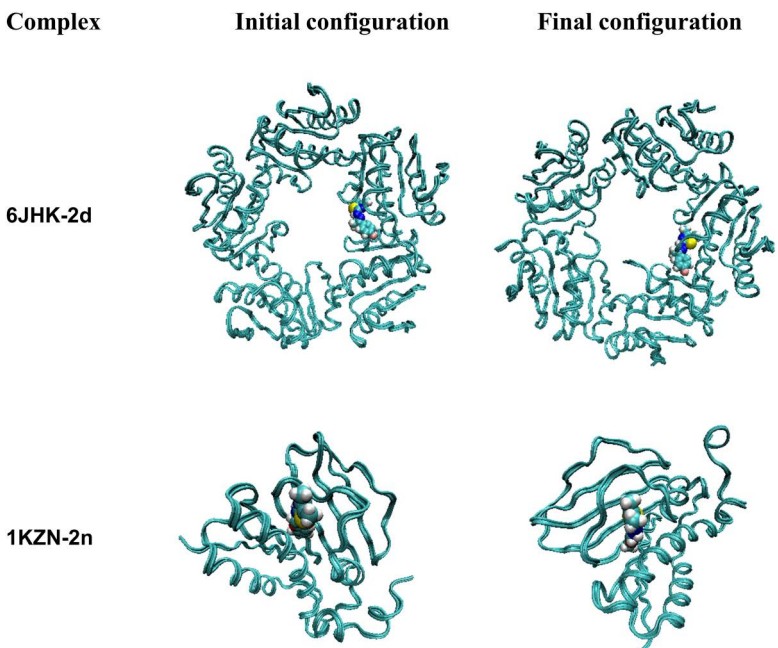

6JHK-2d

1KZN-2n

**Fig 8. Initial and final snapshots of two different complexes during 100 ns MD trajectory.**

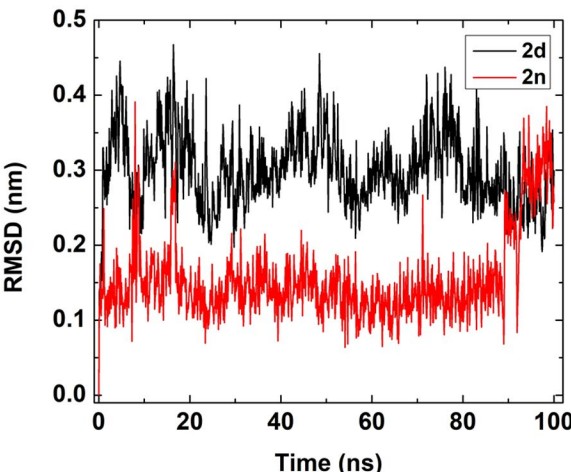

**Fig 9. RMSD plot of two ligands in two different complexes with respect to the equilibrated initial structure.**

binding within the active site of 1KZN. Despite the lower fluctuations observed in the residues within the active site of ligand (**2d**), it does not bind there.

**MM-PBSA binding energies.** The binding energies were computed using both polar and non-polar solvation criteria throughout the last 80 ns of an MD trajectory. The findings presented in **Table 4** pertain to binding ligands (**2d**) and (**2n**) with proteins 6JHK and 1KZN, respectively.

For ligands (**2d**) and (**2n**), Van der Waals energy (vdW-E) predominantly contributed to the overall negative binding energy (BE) values. The maximum contribution was observed for ligand (**2n**) (-153.10 kJ/mol), followed by (**2d**) (-102.05 kJ/mol). Additionally, electrostatic

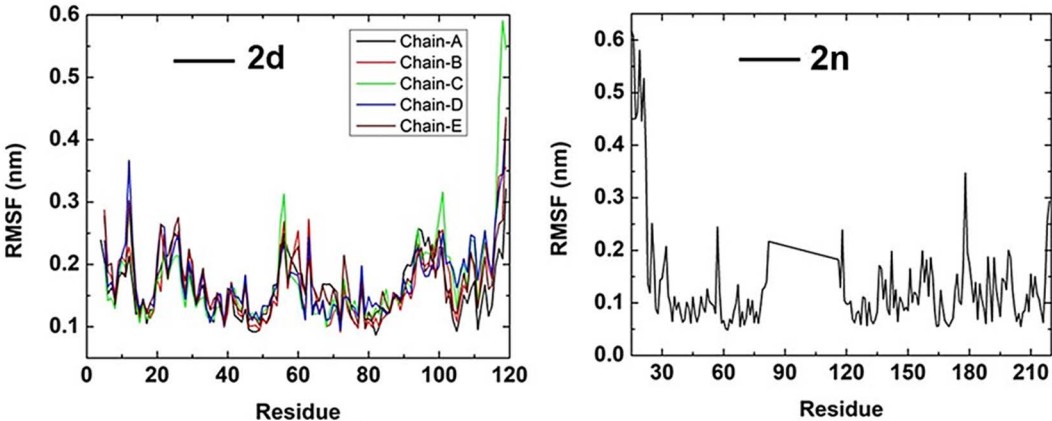

**Fig 10. Root Mean Square Fluctuation (RMSF) analysis for the two protein-ligand systems.**

**Table 4. Binding energy values and individual component energy calculated in kJ/mol with MM-PBSA method for two ligands bound with two different proteins.**

| Complex | vdW-E | ESE | PSE | SASA-E | BE | SD |
|---------|-------|-----|-----|--------|-----|-----|
| **6JHK-2d** | -102.05 | -62.34 | 152.16 | -13.65 | -25.88 | +/-12.60 |
| **1KZN-2n** | -153.10 | -1.27 | 88.91 | -14.09 | -59.01 | +/-14.70 |

energy (ESE) and solvent-accessible surface area energy (SASA-E) made negative contributions to the BE. However, an increase in polar solvation energy (PSE) resulted in a reduction of the total negative binding energy, indicating unfavorable interactions with solvent molecules. In the case of ligand (**2d**), where vdW energy (-102.05 kJ/mol) played the most significant role, the net binding energy was -25.88 kJ/mol. Conversely, for (**2n**), where vdW energy (-153.10 kJ/mol) was the primary contributor, resulting in a maximum binding energy value of -59.01 kJ/mol, which is 33.13 kJ/mol higher than that of (**2d**). The substantial positive value of PSE (152.16 kJ/mol) renders the interaction with (**2d**) unfavorable, contributing to a lower binding energy. However, for (**2n**), the PSE contributes only 88.91 kJ/mol.

## Conclusions

In the present study, eighteen thiazole Schiff base derivatives **2(a–r)** were synthesized *via* two-step reactions in five series with good yields. The chemical structures of synthesized analogs were evaluated by spectral analyses. *In vitro,* antimicrobial activity of the synthesized derivatives against gram-positive and gram-negative bacterial strains revealed that compounds (**2d**) (48.3 ± 0.6 mm) and (**2n**) (45.3 ± 0.6 mm) displayed enhanced activity compared to standards. *In silico* molecular docking studies of the active compounds against target protein receptors revealed that compounds (**2d**) and (**2n**) are tolerable with good binding scores having acceptable binding interactions. However, MD studies confirmed the binding stability of (**2d**) and (**2n**) within their respective target proteins through RMSD and RMSF analyses, where both compounds mostly kept the range below 0.5 nm. Throughout the 100 ns long MD simulations, (**2d**) and (**2n**) compounds remained bound to their target proteins, as evidenced by their binding free energies (-25.88 kJ/mol and -59.01 kJ/mol, respectively). All the compounds showed promising ADME and drug-like characteristics. *In silico* and experimental results demonstrated in this study could be useful for future drug design and development processes. More details on toxicological analysis with substituents and mechanism of action need to be

studied in the future. Green methods would be anticipated in further study with more new derivatives to minimize the solvents and byproducts.

## Supporting information

**S1 Fig.  IR spectrum of 2a.**
(TIF)

**S2 Fig.**  [1]H NMR spectrum of **2a**.
(TIF)

**S3 Fig.  HRMS data of 2a.**
(TIF)

**S4 Fig.  IR spectrum of 2b.**
(TIF)

**S5 Fig.**  [1]H NMR spectrum of **2b**.
(TIF)

**S6 Fig.  HRMS data of 2b.**
(TIF)

**S7 Fig.  IR spectrum of 2c.**
(TIF)

**S8 Fig.**  [1]H NMR spectrum of **2c**.
(TIF)

**S9 Fig.  HRMS data of 2c.**
(TIF)

**S10 Fig.  IR spectrum of 2d.**
(TIF)

**S11 Fig.**  [1]H NMR spectrum of **2d**.
(TIF)

**S12 Fig.  HRMS data of 2d.**
(TIF)

**S13 Fig.  IR spectrum of 2e.**
(TIF)

**S14 Fig.**  [1]H NMR spectrum of **2e**.
(TIF)

**S15 Fig.  HRMS data of 2e.**
(TIF)

**S16 Fig.  IR spectrum of 2f.**
(TIF)

**S17 Fig.**  [1]H NMR spectrum of **2f**.
(TIF)

**S18 Fig.  HRMS data of 2f.**
(TIF)

**S19 Fig.  IR spectrum of 2g.**
(TIF)

**S20 Fig.**  [1]H NMR spectrum of **2g**.
(TIF)

**S21 Fig.  HRMS data of 2g.**
(TIF)

**S22 Fig.  IR spectrum of 2h.**
(TIF)

**S23 Fig.**  [1]H NMR spectrum of **2h**.
(TIF)

**S24 Fig.  HRMS data of 2h.**
(TIF)

**S25 Fig.  IR spectrum of 2i.**
(TIF)

**S26 Fig.**  [1]H NMR spectrum of **2i**.
(TIF)

**S27 Fig.  HRMS data of 2i.**
(TIF)

**S28 Fig.  IR spectrum of 2j.**
(TIF)

**S29 Fig.**  [1]H NMR spectrum of **2j**.
(TIF)

**S30 Fig.  HRMS data of 2j.**
(TIF)

**S31 Fig.  IR spectrum of 2k.**
(TIF)

**S32 Fig.**  [1]H NMR spectrum of **2k**.
(TIF)

**S33 Fig.  HRMS data of 2k.**
(TIF)

**S34 Fig.  IR spectrum of 2l.**
(TIF)

**S35 Fig.**  [1]H NMR spectrum of **2l**.
(TIF)

**S36 Fig.  HRMS data of 2l.**
(TIF)

**S37 Fig.  IR spectrum of 2m.**
(TIF)

**S38 Fig.**  [1]H NMR spectrum of **2m**.
(TIF)

## Acknowledgments

We are indebted to the Instrumental Analysis Center, Kumamoto University, Kumamoto, Japan, for HRMS analyses.

## Author contributions

**Conceptualization:** Mohammad Sayed Alam, Md. Aminul Haque, Mohammad Mostafizur Rahman.

**Data curation:** Tanmoy Kumar Kundu, Ismail Hossain.

**Formal analysis:** Md. Din Islam, Tanmoy Kumar Kundu, Ismail Hossain, Hiroshi Nishino.

**Investigation:** Tanmoy Kumar Kundu, Ismail Hossain.

**Methodology:** Mohammad Sayed Alam, Md. Aminul Haque, Mohammad Mostafizur Rahman.

**Project administration:** Md. Aminul Haque.

**Resources:** Md. Aminul Haque, Mohammad Mostafizur Rahman.

**Software:** Joyanta Kumar Saha, Sumita Saznin Marufa.

**Supervision:** Md. Aminul Haque, Mohammad Mostafizur Rahman.

**Validation:** Md. Din Islam, Md. Aminul Haque, Mohammad Mostafizur Rahman.

**Visualization:** Joyanta Kumar Saha, Sumita Saznin Marufa.

**Writing – original draft:** Md. Din Islam, Mohammad Mostafizur Rahman.

**Writing – review & editing:** Sumita Saznin Marufa, Hiroshi Nishino, Md. Aminul Haque, Mohammad Mostafizur Rahman.

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
