## [Decision Letter · Decision Letter 0]

26 Nov 2024

PONE-D-24-40454Synthesis, antibacterial activity, in silico ADMET prediction, docking, and molecular dynamics studies of aryl and heteroaryl ring containing thiazole Schiff base derivativesPLOS ONE

Dear Dr. Rahman,

Thank you for submitting your manuscript to PLOS ONE. After careful consideration, we feel that it has merit but does not fully meet PLOS ONE’s publication criteria as it currently stands. Therefore, we invite you to submit a revised version of the manuscript that addresses the points raised during the review process.

We look forward to receiving your revised manuscript.

Kind regards,

Wagdy M. Eldehna, Ph.d

Academic Editor

PLOS ONE

Journal Requirements: When submitting your revision, we need you to address these additional requirements. 1. Please ensure that your manuscript meets PLOS ONE's style requirements, including those for file naming. The PLOS ONE style templates can be found at https://journals.plos.org/plosone/s/file?id=wjVg/PLOSOne_formatting_sample_main_body.pdf and https://journals.plos.org/plosone/s/file?id=ba62/PLOSOne_formatting_sample_title_authors_affiliations.pdf 2. Please note that PLOS ONE has spec6ific guidelines on code sharing for submissions in which author-generated code underpins the findings in the manuscript. In these cases, all author-generated code must be made available without restrictions upon publication of the work. Please review our guidelines at https://journals.plos.org/plosone/s/materials-and-software-sharing#loc-sharing-code and ensure that your code is shared in a way that follows best practice and facilitates reproducibility and reuse. 3. We note that this submission includes NMR spectroscopy data. We would recommend that you include the following information in your methods section or as Supporting Information files:  1) The make/source of the NMR instrument used in your study, as well as the magnetic field strength. For each individual experiment, please also list: the nucleus being measured; the sample concentration; the solvent in which the sample is dissolved and if solvent signal suppression was used; the reference standard and the temperature.  2) A list of the chemical shifts for all compounds characterised by NMR spectroscopy, specifying, where relevant: the chemical shift (δ), the multiplicity and the coupling constants (in Hz), for the appropriate nuclei used for assignment.  3)The full integrated NMR spectrum, clearly labelled with the compound name and chemical structure.We also strongly encourage authors to provide primary NMR data files, in particular for new compounds which have not been characterised in the existing literature. Authors should provide the acquisition data, FID files and processing parameters for each experiment, clearly labelled with the compound name and identifier, as well as a structure file for each provided dataset. See our list of recommended repositories here: https://journals.plos.org/plosone/s/recommended-repositories 4. We noticed you have some minor occurrence of overlapping text with the following previous publication(s), which needs to be addressed: -Synthesis, antimicrobial and antioxidant evaluation with in silico studies of new thiazole Schiff base derivatives (https://doi.org/10.1016/j.molstruc.2021.131465) (among others) In your revision ensure you cite all your sources (including your own works), and quote or rephrase any duplicated text outside the methods section. Further consideration is dependent on these concerns being addressed. 5. Thank you for stating the following financial disclosure: "Authors are thankful to the Ministry of Science and Technology (MOST), Bangladesh for funding in this project (Project number PHY’S 553, 2020-2021). The project was partially supported by Jagannath University, Dhaka, Bangladesh." Please state what role the funders took in the study.  If the funders had no role, please state: ""The funders had no role in study design, data collection and analysis, decision to publish, or preparation of the manuscript."" If this statement is not correct you must amend it as needed. Please include this amended Role of Funder statement in your cover letter; we will change the online submission form on your behalf. 6. Thank you for stating in your Funding Statement: "Authors are thankful to the Ministry of Science and Technology (MOST), Bangladesh for funding in this project (Project number PHY’S 553, 2020-2021). The project was partially supported by Jagannath University, Dhaka, Bangladesh." Please provide an amended statement that declares *all* the funding or sources of support (whether external or internal to your organization) received during this study, as detailed online in our guide for authors at http://journals.plos.org/plosone/s/submit-now.  Please also include the statement “There was no additional external funding received for this study.” in your updated Funding Statement. Please include your amended Funding Statement within your cover letter. We will change the online submission form on your behalf.

Reviewers' comments:

Reviewer's Responses to Questions

**Comments to the Author**

1. Is the manuscript technically sound, and do the data support the conclusions?

Reviewer #1: Yes

Reviewer #2: Yes

Reviewer #3: Yes

2. Has the statistical analysis been performed appropriately and rigorously? 

Reviewer #1: No

Reviewer #2: Yes

Reviewer #3: No

3. Have the authors made all data underlying the findings in their manuscript fully available?

Reviewer #1: Yes

Reviewer #2: Yes

Reviewer #3: Yes

4. Is the manuscript presented in an intelligible fashion and written in standard English?

Reviewer #1: Yes

Reviewer #2: Yes

Reviewer #3: No

5. Review Comments to the Author

Reviewer #1: 1. The introduction lacks a detailed discussion on the novelty of the synthesized thiazole Schiff base derivatives. More emphasis is needed on how these compounds stand out from other similar derivatives in terms of structure and biological activity.

2. The rationale behind selecting the specific aryl and heteroaryl derivatives for synthesis is not clearly explained. A brief justification of the choice of functional groups and their expected effects on biological activity would strengthen the manuscript.

3. The discussion on how the thiazole Schiff base derivatives interact with the bacterial cell wall or proteins is not sufficiently detailed.

4. Some synthesized compounds, such as 2e, 2g, and 2h, showed no activity, but there is no thorough discussion of why this might be the case. A comparison between active and inactive compounds in terms of structural differences could help explain these results.

5. The molecular docking section lacks clarity on how the protein receptors (6JHK for B. subtilis and 1KZN for E. coli) were selected. Explaining why these receptors are relevant targets for the synthesized compounds would make the docking studies more compelling.

6. While the manuscript includes DFT calculations for structure optimization, there is no detailed explanation of the relevance of these calculations to the overall study. Clarifying how DFT helps in understanding the molecular properties of the synthesized compounds would be beneficial.

7. The ADMET prediction results are presented in tabular form, but the implications of these results for the potential development of oral drugs are not adequately discussed. More analysis is needed to highlight the importance of ADMET properties like %ABS and toxicity.

8. There is no mention of statistical tests used to analyze the antibacterial data. Applying statistical analysis would add rigor to the comparisons between different compounds and standard antibiotics.

9. Why only this particular basis set has been used for DFT? Improvement is required in the theoretical section. Go through the following recent articles with DFT/docking/MD and support the manuscript with the following references to make it more- 10.1016/j.ijleo.2022.169367; 10.1080/07391102.2022.2162583; 10.1016/j.molliq.2024.124412; https://dx.doi.org/10.4314/bcse.v37i2.19

10. The conclusion is concise but lacks a forward-looking perspective. Expanding on potential future directions, such as exploring the clinical relevance of the synthesized compounds or testing them in more complex biological systems, would strengthen the conclusion.

Reviewer #2: Comments

1) In the abstract, there are terms like ‘conspicuously’. It would be better to use some scientific terms instead of these terms.

2) It would be better to use the compound numbers as 2(a-l) instead of 2a-2l. Wherever applicable, it should be corrected.

3) English and grammar corrections should be done with utmost care. Overall manuscript should be revised in these aspects.

4) References section also should be rechecked according to the journal guidelines.

5) Plagiarism should be checked.

6) Schiff base synthetic approaches are really commendable and results are promising.

7) Manuscript has many merits.

Report

After addressing the above comments, I would strongly recommend to “accept and publish” in “PlosOne’.

Reviewer #3: Respected Editor,

Submitted manuscript entitled "Synthesis, antibacterial activity, in silico ADMET prediction, docking, and molecular dynamics studies of aryl and heteroaryl ring containing thiazole Schiff base derivatives" by Islam et al., seemed interesting. However, given all the points raised below, some issues must be addressed before accepting for publication.

1- I suggest to modify the title.

2- Codes of the synthesized compounds must be in the bracket

3- It is suggested to reduce and duplications, such as Scheme 2 and Scheme 3, can write as Schemes 2 and 3.

4- Abstract section is weak. Please improve it and add problem statement/research gap.

5- The authors should investigate MIC studies for the compounds. Only inhibition zone determinations is not enough for recent studies.

6- How active site of the target receptors are determined? The have co-crystalized ligand, authors should use to validated the reliability of docking procedure.

7- Some references for biological activities, and computational parts are recommended: 10.2174/1877946812666220928102954, https://doi.org/10.1021/acsomega.4c03679, https://doi.org/10.1007/s11164-024-05354-x, https://doi.org/10.1021/acsomega.3c08407

8- The conclusion should contain more of the quantitative data.

9- The whole manuscript needs to be checked to avoid typo-error.

6. PLOS authors have the option to publish the peer review history of their article (what does this mean? ). If published, this will include your full peer review and any attached files.

**Do you want your identity to be public for this peer review?** For information about this choice, including consent withdrawal, please see our Privacy Policy .

Reviewer #1: **Yes: ** Dr. Sonam Shakya

Reviewer #2: No

Reviewer #3: No

---

## [Author Response · Author response to Decision Letter 0]

8 Jan 2025

Response to Reviewers (PONE-D-24-40454)

Response to Reviewer #1

1. The introduction lacks a detailed discussion on the novelty of the synthesized thiazole Schiff base derivatives. More emphasis is needed on how these compounds stand out from other similar derivatives in terms of structure and biological activity.

Response: Several sentences with references were added in introduction section regarding the novelty of the synthesized compounds and highlighted with yellow marks.

2. The rationale behind selecting the specific aryl and heteroaryl derivatives for synthesis is not clearly explained. A brief justification of the choice of functional groups and their expected effects on biological activity would strengthen the manuscript.

Response: Several sentences with references were added in the introduction section regarding the rationale behind selecting the specific aryl and heteroaryl derivatives for synthesizing the new derivatives and highlighted with yellow marks.

3. The discussion on how the thiazole Schiff base derivatives interact with the bacterial cell wall or proteins is not sufficiently detailed.

Response: Several sentences were written in the results and discussion section (Molecular docking studies) with reference to clarify the interaction of the derivatives with the bacterial cell wall and marked in yellow.

4. Some synthesized compounds, such as 2e, 2g, and 2h, showed no activity, but there is no thorough discussion of why this might be the case. A comparison between active and inactive compounds in terms of structural differences could help explain these results.

Response: According to the reviewer's comments, the antimicrobial activity part in results and discussion section was re-written with references to explain the structure-activity relationship and highlighted with yellow color in the revised manuscript.

5. The molecular docking section lacks clarity on how the protein receptors (6JHK for B. subtilis and 1KZN for E. coli) were selected. Explaining why these receptors are relevant targets for the synthesized compounds would make the docking studies more compelling.

Response: Several sentences regarding the docking validation of the ligands against the respective receptors were added and highlighted with yellow color (in molecular docking sub-section of results and discussion section).

6. While the manuscript includes DFT calculations for structure optimization, there is no detailed explanation of the relevance of these calculations to the overall study. Clarifying how DFT helps in understanding the molecular properties of the synthesized compounds would be beneficial.

Response: According to the reviewer’s comment, several sentences were added in the revised manuscript to discuss the relevance of DFT method with references in the Experimental section, which is marked in yellow.

7. The ADMET prediction results are presented in tabular form, but the implications of these results for the potential development of oral drugs are not adequately discussed. More analysis is needed to highlight the importance of ADMET properties like %ABS and toxicity.

Response: The ADMET prediction results were explained adequately to make the tabular data understandable. Moreover, a sentence was added in the results and discussion section (In silico ADMET prediction) to make it clearer and highlighted with yellow mark.

8. There is no mention of statistical tests used to analyze the antibacterial data. Applying statistical analysis would add rigor to the comparisons between different compounds and standard antibiotics.

Response: We used “agar disk diffusion” method to analyze the antibacterial potency of the synthesized molecules and compared them against the standard ceftriaxone, which is represented in Table 1 and discussed in the result and discussion section. All experiments were repeated three times and data were taken as mean ± standard deviation which was mentioned as footnote under the table.

9. Why only this particular basis set has been used for DFT? Improvement is required in the theoretical section. Go through the following recent articles with DFT/docking/MD and support the manuscript with the following references to make it more- 10.1016/j.ijleo.2022.169367; 10.1080/07391102.2022.2162583; 10.1016/j.molliq.2024.124412; https://dx.doi.org/10.4314/bcse.v37i2.19

Response: Several sentences were added in the revised manuscript to discuss the relevance of the DFT method with the above-mentioned references in the experimental section and marked in yellow.

10. The conclusion is concise but lacks a forward-looking perspective. Expanding on potential future directions, such as exploring the clinical relevance of the synthesized compounds or testing them in more complex biological systems, would strengthen the conclusion.

Response: Several sentences regarding the forward-looking perspective were added in the conclusion section and marked in yellow.

Response to Reviewer #2

Comments

1) In the abstract, there are terms like ‘conspicuously’. It would be better to use some scientific terms instead of these terms.

Response: The word ‘conspicuously’ was replaced by the word ‘prominently’ according to the reviewer's comments and marked in yellow.

2) It would be better to use the compound numbers as 2(a-l) instead of 2a-2l. Wherever applicable, it should be corrected.

Response: Compound numbers were re-written in the revised manuscript according to the reviewer's comments and marked them in yellow.

3) English and grammar corrections should be done with utmost care. Overall manuscript should be revised in these aspects.

Response: We have gone through the manuscript and corrected the grammatical and typographical errors in the revised manuscript, where found and highlighted with yellow color.

4) References section also should be rechecked according to the journal guidelines.

Response: We re-checked the references attentively, organized them according to the Journal’s rules, and marked them with yellow.

5) Plagiarism should be checked.

Response: After revise the manuscript, plagiarism was checked by iThenticate software. No significant plagiarism was detected. Mostly spectral data, figure captions, affiliations, title of different sections and sub-sections, references, and some scientific common terms were detected during checking.

Response to Reviewer #3

1- I suggest to modify the title.

Response: The title of the manuscript was modified according to the reviewer’s comment to make it more specific. The term ‘aryl/heteroaryl’ was replaced by ‘substituted phenyl and furan’ in the revised title and marked in yellow.

2- Codes of the synthesized compounds must be in the bracket

Response: Codes of the compounds were re-written in the revised manuscript according to the reviewer's comments and marked in yellow.

3- It is suggested to reduce and duplications, such as Scheme 2 and Scheme 3, can write as Schemes 2 and 3.

Response: It was corrected accordingly and marked them in yellow in the revised manuscript.

4- Abstract section is weak. Please improve it and add problem statement/research gap.

Response: Abstract section was re-written mentioning research gap in the revised manuscript and marked in yellow.

5- The authors should investigate MIC studies for the compounds. Only inhibition zone determinations is not enough for recent studies.

Response: According to reviewer’s comment, the minimal inhibitory concentrations (MICs) of the two most active compounds 2d and 2n were determined against B. subtilis bacterial strain. The values were added in the results and discussion section (Antibacterial activity) and highlighted in yellow.

6- How active site of the target receptors are determined? The have co-crystalized ligand, authors should use to validated the reliability of docking procedure.

Response: We have considered the receptor protein’s calculated structure where the ligand interacted and the obtained structure’s binding site to determine the active sites and validate the docking with the respective proteins.

7- Some references for biological activities, and computational parts are recommended: 10.2174/1877946812666220928102954, https://doi.org/10.1021/acsomega.4c03679, https://doi.org/10.1007/s11164-024-05354-x, https://doi.org/10.1021/acsomega.3c08407

Response: Three of the above-mentioned papers were cited in the results and discussion section (Ref. 50–52) as one was not found, and the other reference numbers were changed accordingly.

8- The conclusion should contain more of the quantitative data.

Response: According to the reviewer's comment, few sentences regarding the quantitative outcomes of the present work were included in the conclusion section and highlighted in yellow.

9- The whole manuscript needs to be checked to avoid typo-error.

Response: We have gone through the manuscript, and linguistic errors were corrected in the revised manuscript where applicable.

---

## [Decision Letter · Decision Letter 1]

26 Jan 2025

Synthesis, antibacterial activity, in silico ADMET prediction, docking, and molecular dynamics studies of substituted phenyl and furan ring containing thiazole Schiff base derivatives

PONE-D-24-40454R1

Dear Dr. Rahman,

We’re pleased to inform you that your manuscript has been judged scientifically suitable for publication and will be formally accepted for publication once it meets all outstanding technical requirements.

Kind regards,

Wagdy M. Eldehna, Ph.d

Academic Editor

PLOS ONE

Additional Editor Comments (optional):

Reviewers' comments:

Reviewer's Responses to Questions

**Comments to the Author**

1. If the authors have adequately addressed your comments raised in a previous round of review and you feel that this manuscript is now acceptable for publication, you may indicate that here to bypass the “Comments to the Author” section, enter your conflict of interest statement in the “Confidential to Editor” section, and submit your "Accept" recommendation.

Reviewer #1: All comments have been addressed

Reviewer #3: (No Response)

2. Is the manuscript technically sound, and do the data support the conclusions?

Reviewer #1: Yes

Reviewer #3: (No Response)

3. Has the statistical analysis been performed appropriately and rigorously? 

Reviewer #1: (No Response)

Reviewer #3: (No Response)

4. Have the authors made all data underlying the findings in their manuscript fully available?

Reviewer #1: Yes

Reviewer #3: (No Response)

5. Is the manuscript presented in an intelligible fashion and written in standard English?

Reviewer #1: Yes

Reviewer #3: (No Response)

6. Review Comments to the Author

Reviewer #1: (No Response)

Reviewer #3: (No Response)

7. PLOS authors have the option to publish the peer review history of their article (what does this mean? ). If published, this will include your full peer review and any attached files.

**Do you want your identity to be public for this peer review?** For information about this choice, including consent withdrawal, please see our Privacy Policy .

Reviewer #1: **Yes: ** Dr. Sonam Shakya

Reviewer #3: No

---

## [Editor Report · Acceptance letter]

PONE-D-24-40454R1

PLOS ONE

Dear Dr. Rahman,

I'm pleased to inform you that your manuscript has been deemed suitable for publication in PLOS ONE. Congratulations! Your manuscript is now being handed over to our production team.

Kind regards,

on behalf of

Dr. Wagdy M. Eldehna

Academic Editor

PLOS ONE